# The Morphological and Anatomical Traits of the Leaf in Representative *Vinca* Species Observed on Indoor- and Outdoor-Grown Plants

**DOI:** 10.3390/plants10040622

**Published:** 2021-03-24

**Authors:** Alexandra Ciorîță, Septimiu Cassian Tripon, Ioan Gabriel Mircea, Dorina Podar, Lucian Barbu-Tudoran, Cristina Mircea, Marcel Pârvu

**Affiliations:** 1Faculty of Biology and Geology, Babeș-Bolyai University, 44 Republicii Street, 400015 Cluj-Napoca, Romania; alexandra.ciorita@itim-cj.ro (A.C.); dorina.podar@ubbcluj.ro (D.P.); cristina.mircea@ubbcluj.ro (C.M.); 2Electron Microscopy Center, Faculty of Biology and Geology, Babeș-Bolyai University, 5-7 Clinicilor Street, 400006 Cluj-Napoca, Romania; septimiu.tripon@itim-cj.ro (S.C.T.); lucian.barbu@itim-cj.ro (L.B.-T.); 3Integrated Electron Microscopy Laboratory, National Institute for Research and Development of Isotopic and Molecular Technologies, 67-103 Donat Street, 400293 Cluj-Napoca, Romania; 4Faculty of Mathematics and Informatics, Babeș-Bolyai University, 1 M. Kogalniceanu Street, 400084 Cluj-Napoca, Romania; mircea@cs.ubbcluj.ro

**Keywords:** *Vinca* leaves, parenchymal tissues, ultrastructure, electron microscopy, epicuticular waxes, cuticle, computational analysis, mesophyll airspace

## Abstract

Morphological and anatomical traits of the *Vinca* leaf were examined using microscopy techniques. Outdoor *Vinca minor* and *V. herbacea* plants and greenhouse cultivated *V. major* and *V. major* var. *variegata* plants had interspecific variations. All *Vinca* species leaves are hypostomatic. However, except for *V. minor* leaf, few stomata were also present on the upper epidermis. *V. minor* leaf had the highest stomatal index and *V. major* had the lowest, while the distribution of trichomes on the upper epidermis was species-specific. Differentiated palisade and spongy parenchyma tissues were present in all *Vinca* species’ leaves. However, *V. minor* and *V. herbacea* leaves had a more organized anatomical aspect, compared to *V. major* and *V. major* var. *variegata* leaves. Additionally, as a novelty, the cellular to intercellular space ratio of the *Vinca* leaf’s mesophyll was revealed herein with the help of computational analysis. Lipid droplets of different sizes and aspects were localized in the spongy parenchyma cells. Ultrastructural characteristics of the cuticle and its epicuticular waxes were described for the first time. Moreover, thick layers of cutin seemed to be characteristic of the outdoor plants only. This could be an adaptation to the unpredictable environmental conditions, but nevertheless, it might influence the chemical composition of plants.

## 1. Introduction

The *Vinca* genus belongs to the Apocynaceae family and comprises three species and one variety: *V. major* L. (bigleaf periwinkle), *V. minor* L. (lesser periwinkle), *V. herbacea* Walds. & Kit., (herbaceous periwinkle), and *V. major* L. var. *variegata* ‘Louden’ (greater periwinkle with white margins) [1]. Other databases complete the list of species with *Vinca difformis* Pourr., *Vinca erecta* Regel & Schmalh., and *Vinca ispartensis* Koyuncu & Eksi [2].

*Vinca* plants are intensely studied for their medical properties [3,4,5] due to the rich alkaloid content [6,7,8,9]. These natural products are produced and stored in the aerial parts of *Vinca* plants [6,10,11], serving as protection against herbivores and pathogens. [12,13]. *V. minor* is one of the most important medicinal species [14] and the sole source of vincamine in nature, one of the few alkaloids with beneficial effects on cells [3,15]. Quantitative and qualitative differences of the chemical composition in *Vinca* extracts are dependent on species and the high content of alkaloids, flavonoids, and phenolic compounds, are in correlation with different pharmacological effects [3,16,17,18,19].

In extract preparation, besides the plant species, many factors influence the variability and diversity of the chemical composition, such as the plant organ and tissue, harvesting period and environmental conditions, or extraction method and the solvent used [3,6,7,8]. Previous results confirmed that the chemical composition varies with the *Vinca* extract, obtained from different organs such as the leaves of *V. minor* [6,7,14,20]; flowers, leaves, and/or roots of *Vinca sardoa* [21]; or from other different species of *Vinca* [4,8].

Undoubtedly, the leaves are very important organs in this matter since other studies showed their direct connection with the alkaloid content. Plant natural products serve in defence mechanisms and are a mixture of substances that often include alkaloids. In *Euphorbia characias* the aqueous latex produced in the leaves is combined with polyphenols and alkaloids [22]. In *Nicotiana stocktonii* [23] and *Thymus quinquecostatus* [24], alkaloids are stored in the glandular trichomes found on the surface of the leaves. In *Ficus carica*, the alkaloids are synthesized in the epidermal cells and secreted on the surface of the leaves by the glandular trichomes [25,26].

Based on similar studies as the aforementioned, the leaves of *Vinca* species are often used for extract preparation due to the fast release of high alkaloid concentration [11,27], but to our knowledge, their morphology and anatomy are insufficiently studied highlighting the scope of this study. The chemical composition of *Vinca* extracts varies to a great extent due to plant photosynthesis [28,29], which is regulated by numerous environmental and internal factors. These factors include morphological and anatomical leaf traits, i.e., shape, size, and the number of stomata, type or number of trichomes, and the weight of epidermal cuticle [30,31].

The objective of this research was to study these morphological and anatomical traits in the *Vinca* leaf and assess their variability in relation to growth conditions. The cuticles’ ultrastructure and architectural organization of the upper and lower epidermises in the *Vinca* leaves were aimed at each studied species. Moreover, the mesophyll airspace was determined for the first time for the *Vinca* leaves through computational estimation and the anatomy and ultrastructure varied with the species and growth conditions. If these characteristics are responsible for the variability and diversity of the chemical composition of *Vinca* plant extracts, the results of this study could provide an additional dimension to the existing data, with implicit ramifications for extract preparation conditions.

## 2. Results

### 2.1. Vinca Leaf Epidermises

The epidermises of leaves were studied by scanning electron microscopy (SEM) and details such as the presence or absence, type, shape, and distribution of trichomes and stomata were observed.

Pavement cells on the adaxial side of *V. minor*, *V. major*, and *V. major* var. *variegata* leaves had similar typical jigsaw puzzle shapes (Figure 1a,c,e), whereas pavement cells of *V. herbacea* were papillary shaped (Figure 1g). Except for *V. minor*, all species had stomata on the upper epidermis. Tector trichomes with various ornamentations were present on the adaxial leaf side of the leaves as well (Figure 1b,d,f,h).

On the abaxial leaf side, all *Vinca* species presented typical puzzle-shaped pavement cells, while stomata were randomly distributed on the entire surface, except for the midvein (Figure 2).

### 2.2. Vinca Trichomes and Stomata Characteristics

Unicellular conical trichomes were observed on the margins of the adaxial side of *V. major*, *V. major* var. *variegata*, and *V. herbacea* leaves (Figure 3), while *V. minor* had trichomes only on the midvein. The trichomes had the same orientation in all *Vinca* species, i.e., from the petiole to the apex (from back to front).

The morphological parameters of the stomata and trichomes, and their distribution were calculated for each side of the leaves (Table 1).

Trichomes have protective roles in plants and their number is usually regulated by the same factors that affect the number of stomata [32]. As the *Vinca* trichomes are non-glandular, their main role remains as protectors against water loss, and their density is inversely correlated (Figure 4) with the stomatal index (r = −0.7).

### 2.3. Vinca Mesophyll

The semithin sections (Figure 5) helped to determine the anatomical features of the *Vinca* leaves (Table 2). The mesophyll of *V. minor* and *V. herbacea* had a similar structure, with obvious delimitation between the palisade and spongy parenchyma (Figure 5a,d). In *V. major* and *V. major* var. *variegata*, the palisade parenchyma had a lax aspect and large intercellular spaces (Figure 5b,c).

A computational estimation of the intercellular space in the mesophyll was conducted (Figure 6) revealing a high percentage (over 60%) of airspaces for the studied *Vinca* species.

### 2.4. Vinca Epidermal Cell Walls and Cuticle Layers

For a thorough investigation of the anatomical features of *Vinca* leaves, transmission electron microscopy (TEM) analyses were performed. The external pavement cell walls in the upper and lower epidermises (Appendix A) were investigated (Figure 7). It was observed that except for *V. major* (Figure 7c,d), all leaves had a thin layer of epicuticular waxes present above the cutin layers.

The thickness of the upper and lower epidermises was measured with the help of TEM images. The results showed that the upper epidermis is significantly thicker than the lower epidermis for *V. minor* (*p* = 0.005), *V. major* var. *variegata* (*p* = 0.03), and *V. herbacea* (*p* = 0.0002), and of all species, *V. major* had the thinnest cell walls on both epidermises (Figure 8a). Also, the cuticle present on the upper epidermis was thicker than the cuticle on the lower epidermis for *V. minor* and *V. herbacea* (Figure 8b).

Large lipid droplets were found inside the spongy parenchyma cells in leaves of *V. minor*, *V. major*, and *V. major* var. *variegata* as shown by the TEM micrographs (Figure 9a–c). No lipid droplets were observed in *V. herbacea* (Figure 9d).

## 3. Discussion

There are quantitative and qualitative differences regarding the chemical composition of *Vinca* extracts, and previous studies showed that the morphological and anatomical traits of the leaves (shape and surface, cuticle, stomata, trichome, epidermis, etc.) are responsible for the chemical composition variability in the extracts [33,34,35,36]. These morphological and anatomical aspects of the *Vinca* leaf had been investigated in this study as well. The leaf surface area and dry leaf mass are directly correlated for vines species (*Vinca major*, *Trachelospermum jasminoides*, *Hedera nepalensis* var. *sinensis*) [37] and for *Arabidopsis thaliana* [38]. This aspect is important in extract preparation for maximal exploitation of the plants.

Additionally, the morphological characters of all plant organs are used for accurate identification of species [14,33,34,35]. The leaf shape varies within species and populations, and even within the same individual plants; therefore, it affects photosynthesis, water balance, temperature control, and the interactions with other organisms [39].

On the same surface of the *Vinca* leaf within different species, the epidermal cells are distinct, and here was shown that they could vary on both sides even for the same species (Figure 1, Figure 2 and Figure 3). The *Vinca* leaf epidermis has a single layer of cells composed of pavement cells, stomatal complexes, and trichomes (Figure 1, Figure 2, Figure 3, Figure 4 and Figure 5), that serves as a protective barrier to environmental factors [40]. On the adaxial side, *Vinca* leaves had puzzle-shaped epidermal cells, except for *V. herbacea* that had papillary epidermal cells. Similar results were previously reported for *V. minor* and *V. herbacea* only [41], and there are several theories that try to explain this type of organization of the pavement cells [42,43,44].

These specialized epidermal cells differ intra- and interspecific, depending on various extrinsic and intrinsic parameters, e.g., the leaf age, dimension, insertion position on the stem, combined with CO_2_ atmospheric concentration, irradiance, humidity, and other environmental factors (pollution, pathogens, etc.) [44,45,46]. Therefore, the current study was conducted under minimum variation of these parameters. The outdoor plants (*V. minor* and *V. herbacea*) grew in a shaded area and had the same light exposition (46°45′51” N; 23°34′47” E). The indoor plants (*V. major* and *V. major* var. *variegata*) were exposed only to sunlight with the same circadian cycles as the outdoor plants and the humidity and CO_2_ levels were regulated so that they could resemble the natural conditions. All the above-mentioned criteria play vital roles for the medicinal taxa [47,48].

Also, a connection between the cell walls of the leaves’ epidermis and the natural products in plants had been previously established [6]. In most plant species, the epidermis has specialized roles in the biosynthesis and accumulation of a wide range of natural products, including alkaloids [40], terpenes, and flavonoids [40,49], indicating that the variances observed here could be indicative of the chemical variation of the plant extracts.

The *Vinca* leaf epidermal cells are covered by a cuticle (Figure 5, Figure 7 and Figure 8) with complex chemical composition and have different thicknesses on the upper and lower epidermises as previously indicated [50,51,52]. In terms of electron-dense cutin layers, the *V. minor*, *V. major* var. *variegata* and *V. herbacea* leaves are similar to the leaves of *Pyrus communis* and *Populus bolleana* [50], but were never described for the *Vinca* species. The cuticular waxes play significant roles in *Vinca* species due to their involvement in the defense mechanisms against pathogens, temperature variations, salinity, or excessive ultraviolet radiations [52].

Glandular and secretory trichomes are responsible for biosynthesis, secretion and/or accumulation of phytochemicals useful for defense [40]. In the leaves of *Vinca* species analyzed herein, only tector (protective) trichomes had been identified, with the main role in defense mechanisms against water loss. The number of trichomes increases while the stomatal index decreases, and this way, the transpiration process is balanced [53], explaining our results where the stomatal index was found to be inversely proportional to trichomes’ density. For *V. minor*, the trichomes were placed only on the midvein, while the other species had trichomes on the margins and on secondary veins. Similar results had been previously reported as well for *V. minor* and *V. herbacea* [41,54].

Stomata are important structures involved in exchange processes of the plant with the environment, balancing the CO_2_ influx during photosynthesis and water vapor efflux during transpiration [28,33,35,40,55]. Stomata exhibit a diverse range of shapes, sizes, and numbers across different plant species [53]. Leaf stomata in plants from the same genus or from different plants of the same species are variable [56,57]. Significant negative correlation was documented between stomatal density and stomatal shape parameters (stomatal area, stomatal perimeter, stomatal long axis, stomatal short axis) in other plant species as well [58].

The stomatal index is the recommended parameter to be measured since it reports the stomata density in relation to the number and size of the pavement cells, in each area [28,46]. The stomatal index for the adaxial side (around 4%), is evidently lower than the one registered for the abaxial side (20–30%); therefore, the studied *Vinca* species are all hypostomatic.

Our results showed that the overall thickness of the *Vinca* leaf differs with the species and this influences the photosynthetic rates as well [59]. Besides this, the stomatal index has different values for each *Vinca* species, and the chemical composition could vary as well. This is because a higher specific stomatal index is related to a higher net rate of photosynthesis, and consequently, the plant could enhance the carbon fixation [28,32]. For example, several sugars were shown to affect the growth and development of *V. minor* and *C. roseus* leaves [14]. As carbon is present in many sugars and alkaloids, this could also influence the chemical composition in the long run [28].

*Vinca* leaf mesophyll has different thicknesses, a parameter which is species depended [36]. The *Vinca* leaves contain different numbers of columnar palisade cells. It was confirmed that sunlight-grown plants have more columnar palisade cells than those of shade-grown plants [29,34,35]. Similar to our results where the indoor plants had a less differentiated palisade tissue than the outdoor plants, other studies showed that the palisade tissue cells in *V. major* var. *variegata* leaves were single-layered for the plants located at roof level and double-layered for ground-level plants [35]. Another study demonstrated that depending on light exposition, the leaf of *V. minor* had a double-layered palisade tissue and *V. herbacea* leaf, a single-layered one [41]. The shape of palisade cells and the movement of chloroplast in accordance with the light conditions, are essential for efficient leaf photosynthesis, by cell development regulation with the help of phototropin [29]. The number and shape of palisade cells, which compose the photosynthesis unit area, are different for each *Vinca* leaf analyzed herein, therefore the photosynthesis process varies along with the anatomical traits [29] and this might influence the chemical composition as well if the plants are to be considered for extract preparation.

The thickness of the spongy parenchyma in the *Vinca* leaf varies with the species and contains several layers of ovoid, oblong, or circular shaped cells and the mesophyll airspace [34,35,41]. The cells of the spongy parenchyma store important nutrients for the plant [60]; therefore, its thickness would be directly related to the abundance of natural compounds [61]. Other studies described 7 to 8 layers of cells in the spongy tissues of *V. minor* and *V. herbacea* [34,41], while for *V. major* var. *variegata,* the number of layers decreased to 6–7 for plants grown at ground level [35]. The results presented herein showed that the plants grown outdoor had 7 to 8 layers, while the plants grown indoor, had 5 to 6 layers of cells in the spongy parenchyma—results which are consistent with other findings—and this indicates that growth conditions are important parameters that should be considered before extract preparation.

The programmatic method used for computation helped to determine the ratio between intercellular spaces and cells in the mesophyll of the *Vinca* species. First, it must be stressed that the main objective was the accuracy of the mathematical computations with respect to the ground truth. Initially, various methods were employed for the semi-automatic segmentation of the regions in the semithin sections, using state-of-the-art methods such as those based on computer vision [62,63] or machine learning approaches [64,65]. The employed methods obtained the same accuracy levels as those found in the literature.

However, the visual assessment of the results leads to the conclusion that since the end goal was to compute a percentage as close to reality as possible through an automated method of visual segmentation and labeling of the cellular and inter-cellular regions, significant errors in the computational process occurred. In other words, an accuracy level that is acceptable from a computer science perspective may, however, damage the biological scientific accuracy of the research. *V. minor* had the lowest mesophyll airspace (62.8%), followed by *V. major* (63.8%) and very close to each other were *V. major* var. *variegata* and *V. herbacea* (67.3% and 67.2%, respectively). According to recent studies, there is a connection between the mesophyll airspace and functional stomata [36,66], implying that larger airspaces are connected to a higher stomatal index. Since the *Vinca* species are hypostomatic, the high percentage for intercellular spaces is justified in relation to the high stomatal index observed.

The ultrastructural analysis of the leaves offered a more detailed view of the anatomy and confirmed some of the hypotheses regarding the spongy parenchyma. Lipids are used as substitutes for energy production and their presence as big droplets in the *V. major* and *V. major* var. *variegata* leaf could indicate a compensatory mechanism adopted by the plant because of the greenhouse storage conditions [67]. As indicated in other studies, alkaloids are harmful to the plant and are therefore stored or synthesized in small amounts [61] and only when needed, and the chemical composition variability of *Vinca* plant extracts could be reflected by different amounts of lipid droplets identified in leaf spongy parenchyma [67]. Overall, the methods used to determine the morphological and anatomical aspects of the *Vinca* leaves are correlated with each other and can be used to analyze the leaves of other species as well. This is an important step since *Vinca* species are recognized for their medicinal properties, and a brief morphological and anatomical description of a batch could help determine if the plants are suited for extraction.

## 4. Materials and Methods

### 4.1. Plant Material and Growth Conditions

The *Vinca* plants (Figure 10) were collected in April–May from Alexandru Borza Botanical Garden, of Babeş-Bolyai University in Cluj-Napoca. A voucher specimen for each species was deposited at the Herbarium of Babeş-Bolyai University (CL 665977 for *V. minor*, CL 668019 for *V. major*, CL 668018 for *V. major* L. var. *variegata* ‘Louden’, and CL 668021 for *V. herbacea*).

Fully developed leaves from adult plants were collected from approximately the mid part of the stem. *V. minor* and *V. herbacea* grew in the shade within the grounds of the Botanical Garden, while *V. major* and *V. major* var. *variegata* were potted plants grown in the greenhouse. Before harvest, the selected leaves were thoroughly washed with distilled water.

### 4.2. Scanning Electron Microscopy

An optimized and adapted method [68] was used for electron microscopy analyses. Shortly after harvest, the leaves were placed in 2.7% glutaraldehyde (GT) solution in 0.1 M PBS (phosphate buffer saline) at pH 7.2. After 1 h, GT was changed and left for an additional hour, after which the samples were thoroughly washed four times with PBS. For SEM, the leaves were cut in half and further dehydrated using increasing concentrations of acetone as follows: 30 min at 30%, 30 min at 50%, 30 min at 70%, 1 h at 80%, 1 h at 90%, 4 h at 100%, changing the solution every 1 h. Subsequently, the samples were placed in hexamethyldisilazane:acetone solution (1:2, 1:1, 2:1, 1:0 for 1 h each). All steps were conducted at 4 °C and the reagents were acquired from Sigma Aldrich (Merck, Bucharest, Romania). For examination, SEM HITACHI SU8230 (Hitachi, Tokyo, Japan) was used at an acceleration voltage of 30 kV after the samples were covered with a 9-nm- thick layer of gold, using the Quorum Q150T ES turbomolecular pumped coater (Quorum Technologies, London, UK). Each parameter was measured three times and form one leaf. At least six independent measurements were taken in six different spots on the leaf.

### 4.3. Leaf Stomatal Traits

The obtained SEM images were also used to calculate the stomatal index (SI) according to Equation (1), and the stomatal pore index (SPI), according to Equation (2), for each side of the leaves [45,69]. The density of the trichomes per mm^2^ was determined as well.
SI (%) = SD/(SD + E) × 100(1)
where SD is the stomatal density (number of stomata/mm^2^ leaf surface) and E is the epidermal cell density in the same surface area.
SPI (%) = SD × SL^2^ × 10^−4^(2)
where SL is the stomatal pore length calculated for six random stomata present on the surfaces, and the mean was calculated.

### 4.4. Light Microscopy

For light microscopy (LM) and transmission electron microscopy (TEM) analyses, the samples were cut into ~3 mm × 1 mm × 0.5 mm pieces and placed in osmium tetroxide solution (2% OsO_4_ in 0.15 M PBS of 7.4 pH) for 3 h. Afterward, the samples were washed in PBS and then dehydrated with increasing concentrations of acetone, as follows: 15 min at 30%, 15 min at 50%, 15 min at 70%, 30 min at 80 and 90%, 2 h at 100%, changing the solution once every 30 min. Samples were then infiltrated with Epon 812:acetone solution (1:2, 1:1, 2:1, 1:0, 1 h each step) and then left overnight in freshly prepared Epon 812 epoxy resin. The samples were encapsulated in a horizontal silicone matrix at 60 °C for three days. Using glass knives, a Diatome diamond knife (DiATOME, Hatfield, PA, USA), and Leica UC7 ultramicrotome (Leica Microsystems, Wetzlar, Germany), semithin and ultrathin sections were obtained. The semithin sections (300–400 nm thickness) were placed on glass slides and stained with Epoxy tissue stain, after which they were examined using the Olympus BX51 light microscope coupled with a CCD camera (Olympus, Hamburg, Germany).

### 4.5. Leaf Anatomical Traits

Using the semithin sections, the cell tense ration (CTR) and spongy tissue ratio (SR) were calculated, according to Equations (3) and (4) [69]:CTR (%) = PTT/LT × 100(3)
where PT is the palisade tissue thickness, and LT is the leaf thickness, both parameters measured along the leaves’ sections in three different spots, and the mean was calculated.
SR (%) = STT/LT × 100(4)
where STT is the spongy tissue thickness. The parameter was calculated along the leaves’ sections in three different spots, and the mean was calculated.

### 4.6. Transmission Electron Microscopy

The ultrathin sections (80–150 nm thickness) were placed on 200 mesh copper grids and double-stained with uranyl acetate (2.6 g in 20 mL of 50% ethanol in ultrapure water, for 6 min) and lead citrate (1.41 g in 42 mL MQ water and 8 mL NaOH 1 N, pH 12, for 3 min). The samples were examined using the TEM Jeol JEM 1010 (JEOL, Tokyo, Japan), operated at 80 kV, and coupled with a Mega View III digital camera.

### 4.7. Computational Estimation of the Intercellular Space in the Mesophyll

Using the semithin sections obtained, the cellular and intercellular regions were calculated. The determination and labeling were conducted manually, by use of a professional image processing tool (Photoshop, version 13.0; Adobe Systems Inc., San Jose, CA, USA) for the manual segmentation of the regions. Each region of interest was colored differently as follows: the cellular region was labeled green, the intercellular region was labeled grey, the epidermises were labeled blue, and unimportant regions were labeled yellow. The epidermises were excluded from the calculation, as they are not part of the mesophyll.

Further, the images were used as input for a straightforward Python script (version 3.8.6; available online) that merely quantifies the pixels of different colors and later on computes the required surface ratios, based on simple pixel quantization as the ratio between the number of pixels labeled grey over the number of both grey and green pixels.

### 4.8. Statistical Analyses

One-way ANOVA and Tukey statistical analyses were performed using OriginPro 2016 software (OriginLab Corporation, Northampton, MA, USA). At least three independent measurements were conducted for each parameter, and the mean was calculated ± standard error of the mean. The obtained values at significance levels of *p* ≤ 0.05 were considered statistically significant.

## 5. Conclusions

A scan through the leaf morphological and anatomical traits of *Vinca minor*, *V. herbacea*, *V. major* and *V. major* var. *variegata* species revealed complex details. The stomatal index is related to trichome density, cuticle, spongy and palisade parenchyma thickness, and probably to the lipid and alkaloid content as well. As a significant novelty, the epicuticular waxes and cuticles of *Vinca* leaf epidermises were described for the first time, along with a mesophyll airspace determination method. These results provide a new perspective for other plant species as the parameters analyzed herein can directly influence the photosynthetic rates. Because the leaves of various medicinal plants are often used in extract preparation, a thorough and beforehand examination of these morphological and anatomical parameters could help determine if those plants are suited for pharmacological assessments. The investigated traits justify the differences in the chemical composition of *Vinca* extracts as previously observed in other studies as well. Since photosynthesis is responsible for carbon fixation and plant nutrition overall, the morpho-anatomical data of the *Vinca* species complete the literature data while it brings novelty to this field.

## Figures and Tables

**Figure 1 plants-10-00622-f001:**
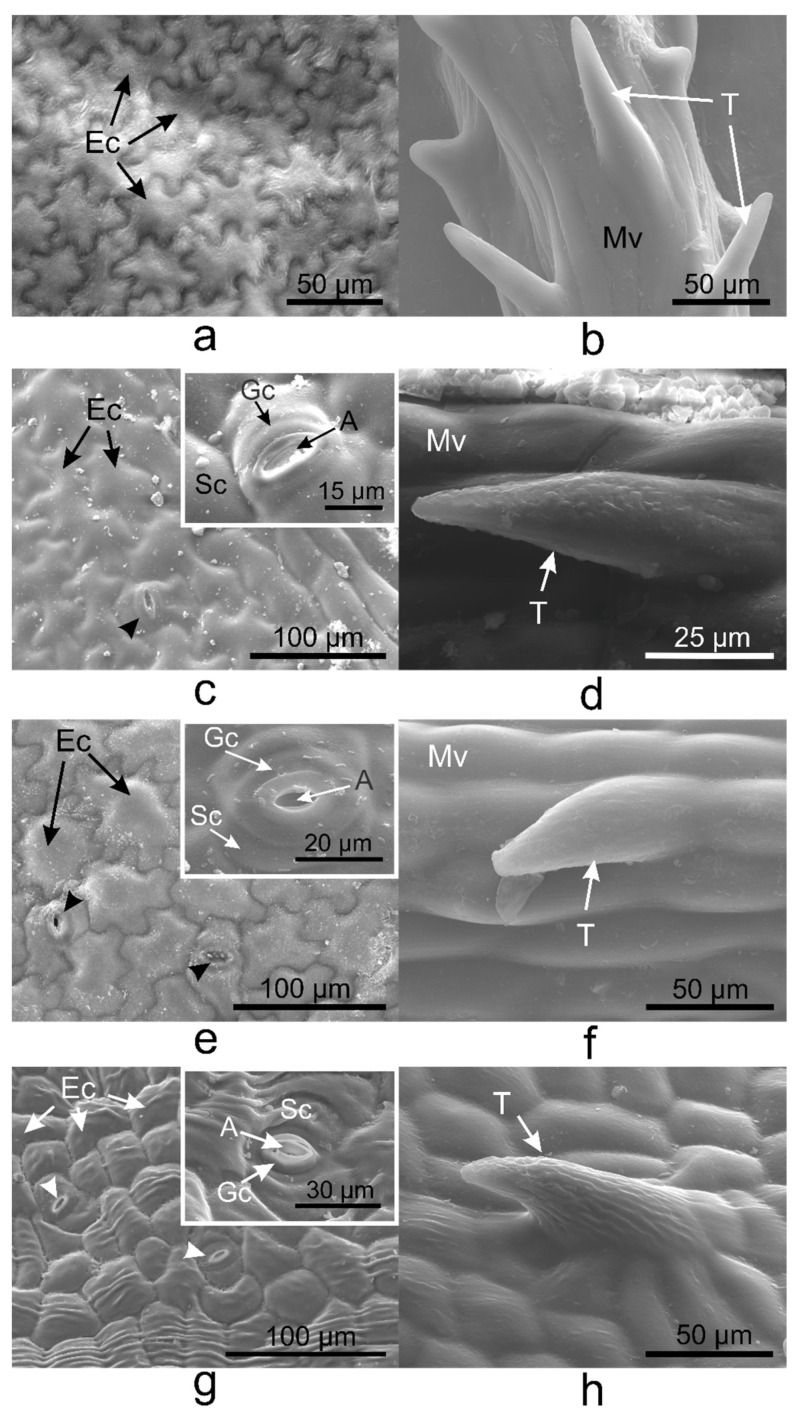
Scanning electron microscopy (SEM) micrographs showing the adaxial (upper) epidermis of the *Vinca* leaves with an overview of the differently shaped epidermal cells and insets with a detailed view of stomata, indicated by the arrowheads (**a**,**c**,**e**,**g**), and non-glandular trichomes observed on the midveins (**b**,**d**,**f**) and secondary veins (**h**). (**a**,**b**) *V. minor*, (**c**,**d**) *V. major*, (**e**,**f**) *V. major* var. *variegata*, (**g**,**h**) *V. herbacea*; A—aperture, Ec—epidermal cell, Gc—guard cell, Mv—midvein, Sc—subsidiary cell, T—trichome.

**Figure 2 plants-10-00622-f002:**
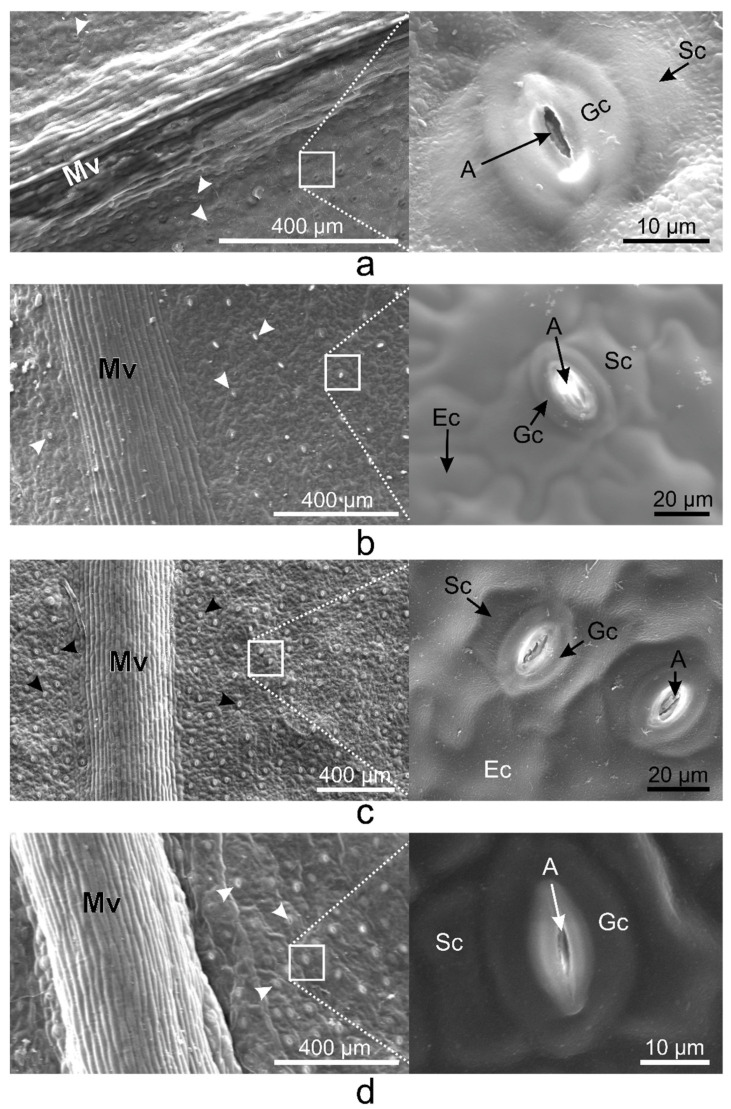
SEM micrographs showing the abaxial (lower) epidermis of the *Vinca* leaves with an overview along the midvein and detailed view of the stomata. (**a**) *V. minor*, (**b**) *V. major*, (**c**) *V. major* var. *variegata*, (**d**) *V. herbacea*; A—aperture, Ec—epidermal cell, Gc—guard cell, Mv—midvein, Sc—subsidiary cell, arrowheads—stomata.

**Figure 3 plants-10-00622-f003:**
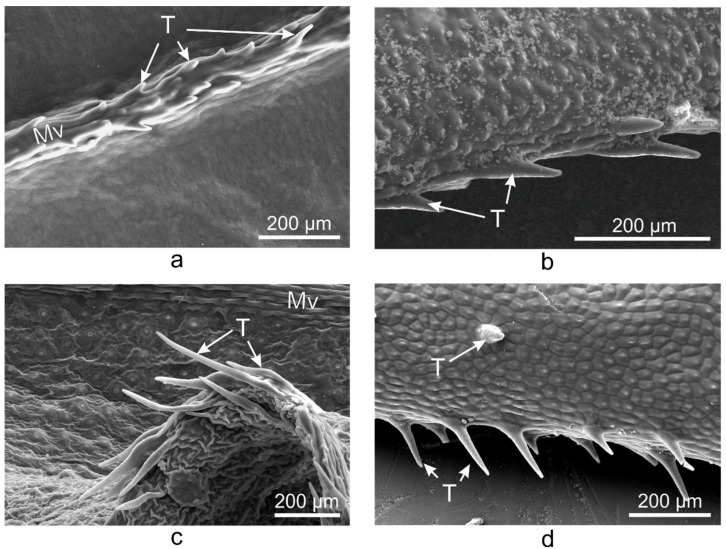
SEM micrographs showing the trichomes present on the midvein of *V. minor* leaf (**a**) and the marginal trichomes on the leaves of *V. major* (**b**), *V. major* var. *variegata* (**c**), and *V. herbacea* (**d**), all oriented towards the apex of the leaf; Mv—midvein, T—trichome.

**Figure 4 plants-10-00622-f004:**
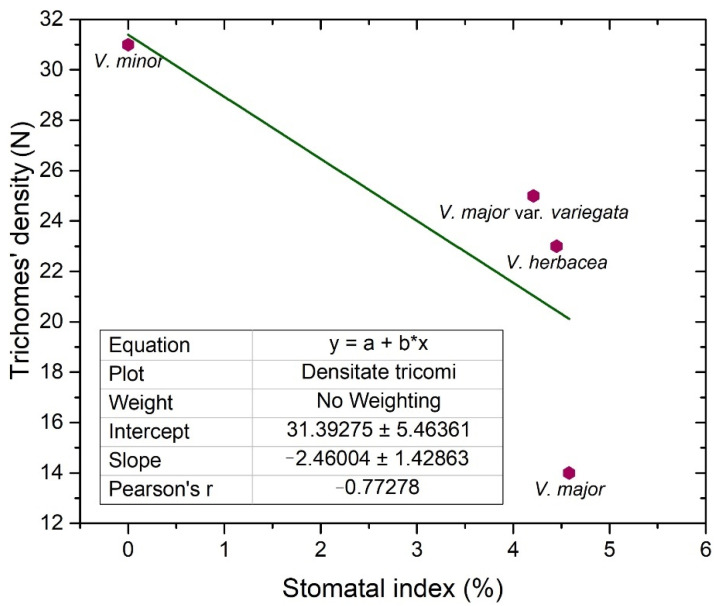
Linear fit of the trichomes’ density and stomatal index on the adaxial side of the examined *Vinca* leaves.

**Figure 5 plants-10-00622-f005:**
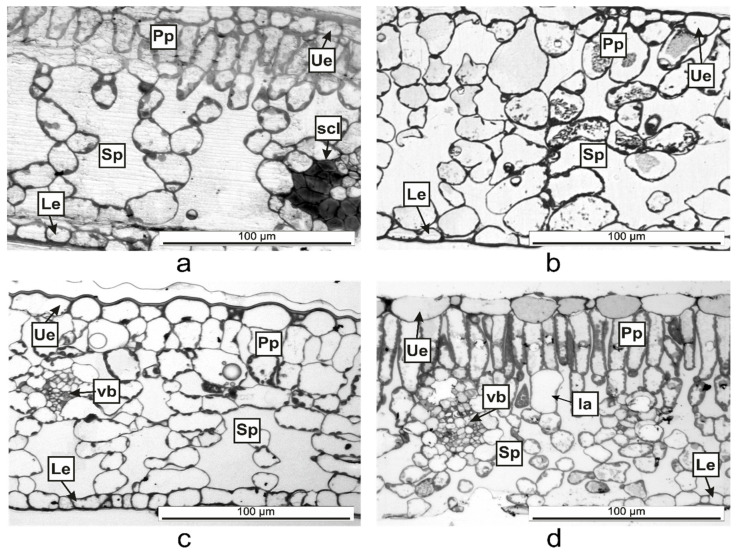
Semithin cross-sections of *Vinca* leaves showing the organization of the mesophyll between the upper epidermis (Ue) and lower epidermis (Le) as determined through light microscopy. (**a**) *V. minor*, (**b**) *V. major*, (**c**) *V. major* var. *variegata*, (**d**) *V. herbacea*; Ia—idioblast, Pp = palisade parenchyma, scl—sclerenchyma, Sp = spongy parenchyma, vb = vascular bundle.

**Figure 6 plants-10-00622-f006:**
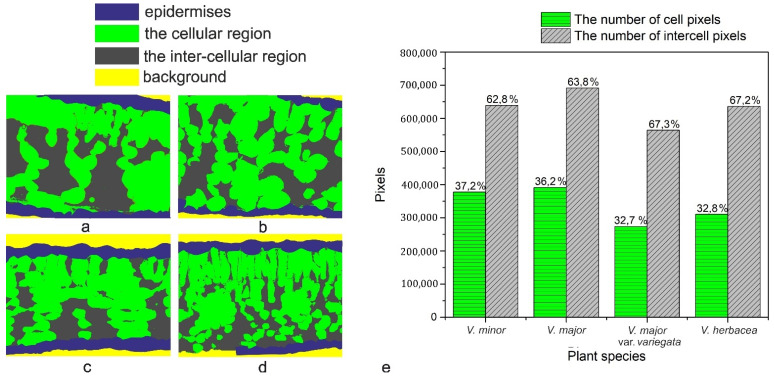
Computational estimation of the intercellular space of the mesophyll. (**a**) *V. minor*, (**b**) *V. major*, (**c**) *V. major* var. *variegata*, (**d**) *V. herbacea*, (**e**) graphical representation of the estimated number of cell pixels and inter-cell pixels as calculated using the Python script.

**Figure 7 plants-10-00622-f007:**
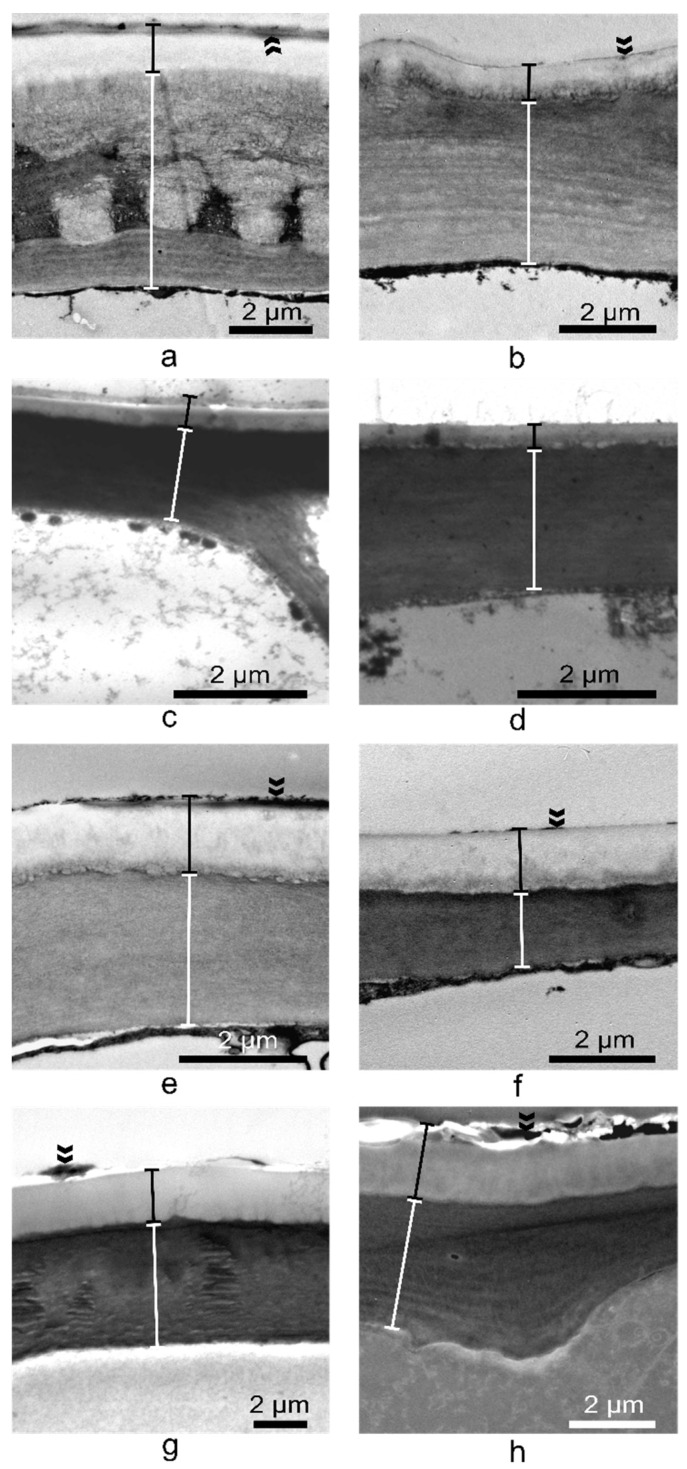
Transmission electron microscopy (TEM) micrographs of the ultrastructure of the upper (**a**,**c**,**e**,**g**) and lower (**b**,**d**,**f**,**h**) epidermises of *Vinca* leaves. (**a**,**b**) *V. minor*, (**c**,**d**) *V. major*, (**e**,**f**) *V. major* var. *variegata*, (**g**,**h**) *V. herbacea*; white delimitations represent the cell wall of the epidermal cells, and black delimitations represent the cuticular layers; double arrowheads indicate the epicuticular waxes.

**Figure 8 plants-10-00622-f008:**
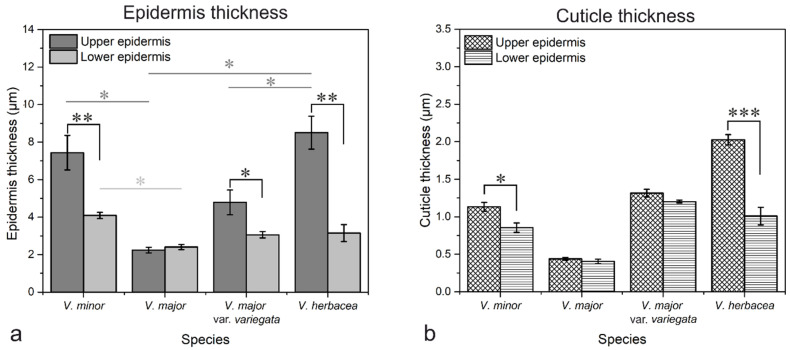
Epidermis (**a**) and cuticle (**b**) measurements showing the differences between the leaves of *Vinca* plants; each value represents the calculated mean of six independent measurements effectuated along the cell walls and cuticles of at least two different cells *** *p* < 0.0001, ** *p* ≤ 0.005, * *p* ≤ 0.05.

**Figure 9 plants-10-00622-f009:**
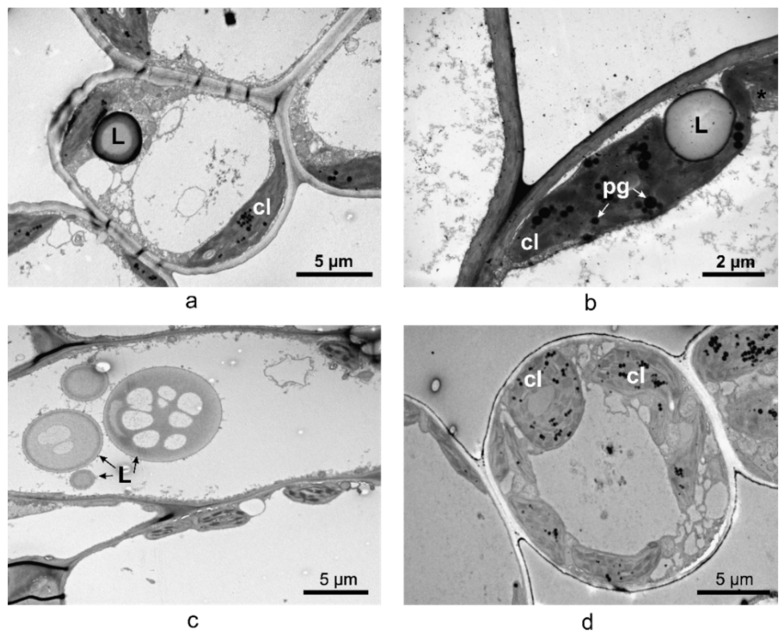
TEM micrographs of the cells located in the spongy parenchyma of (**a**) *V. minor*, (**b**) *V. major*, (**c**) *V. major* var. *variegata*, and (**d**) *V. herbacea* leaves; cl—chloroplast, L—lipid droplets, pg—plastoglobules, *—mitochondrion.

**Figure 10 plants-10-00622-f010:**
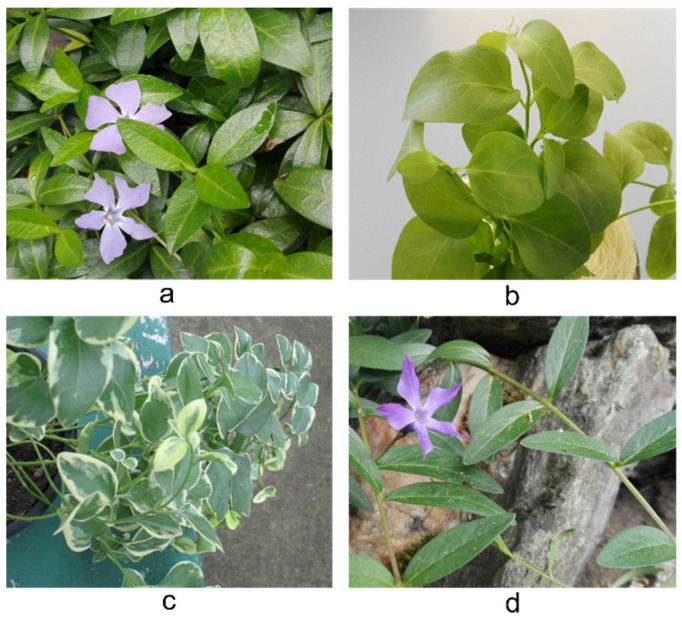
The aerial parts of the *Vinca* plants: (**a**) *V. minor*, (**b**) *V. major*, (**c**) *V. major* var. *variegata*, and (**d**) *V. herbacea.*

**Table 1 plants-10-00622-t001:** Morphological parameters measured on the adaxial and abaxial sides of the *Vinca* leaves.

Species	SD *		SI (%) *		SL (µm)		SPI (%)		TD *
	Ad	Ab	Ad	Ab	Ad	Ab	Ad	Ab	Ad
*V. minor*	0	223	0	25.3	0	9.57 ± 0.6	0	2.04	31
*V. major*	10	49	4.5	23.2	14.84 ± 1.1	19.75 ± 0.4	0.22	1.91	14
*V. major* var. *variegata*	12	68	4.2	20.9	12.33 ± 1	12.24 ± 1.4	0.18	1.01	25
*V. herbacea*	16	60	4.4	25	12.56 ± 1	13.05 ± 1.2	0.25	1.02	23

Ad = adaxial (upper) side, Ab = abaxial (lower) side, SD = Stomatal density, SI (%) = stomatal index, SL = Stomatal pore length, SPI (%) = Stomatal pore index, TD = Trichomes’ density; the criteria marked with * are measured in a one mm^2^ surface area; SL was measured three times on a minimum of six independent stomata ± standard error of the mean (s.e.m.).

**Table 2 plants-10-00622-t002:** The anatomical features measured on the *Vinca* leaves.

Species	LT (µm)	PTT (µm)	STT (µm)	CTR (%)	SR (%)
*V. minor*	85.47 ± 2.6	12.58 ± 0.7	52 ± 0.5	14.72	60.84
*V. major*	136.85 ± 1.4	0	124 ± 0.5	0	90.61
*V. major* var. *variegata*	97.09 ± 1.5	14.57 ± 4.4	67.29 ± 1.7	15.01	69.31
*V. herbacea*	133.38 ± 2.2	36.66 ± 2.2	68.42 ± 3.7	27.48	51.3

LT = Leaf thickness, PTT = Palisade tissue thickness, STT = Spongy tissue thickness, CTR (%) = Cell tense ratio, SR (%) = Spongy tissue ratio; at least three independent measurements were conducted for each parameter measured on the leaf; for LT and STT *n* = 3, ± s.e.m.; for PTT *n* = 6, ± s.e.m.

## Data Availability

Data is available upon appropriate request from the corresponding author.

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
