# Peer review of "The Morphological and Anatomical Traits of the Leaf in Representative Vinca Species Observed on Indoor- and Outdoor-Grown Plants"

_plants, 2021, doi:10.3390/plants10040622_

Round 1

Reviewer 1 Report

The revised manuscript is a study based on the morphological and anatomical analysis of the Vinca species' leaf.

The descriptions are better than the earlier manuscript. I have no difficulty in recommending acceptance.

Author Response

Response to Reviewer 1 Comments

Thank you for your time and support!

Reviewer 2 Report

row 113: area, not aria
row 391 ''meets the eye'' is more journalistic phrase, not common in scientific communication; consider to change it, or omit it. However, the conclusions still could be  (could be!) improved, although are better than during the first round of reviewing.

Author Response

Response to Reviewer 2 Comments

Thank you for your time and for the provided indications. The corrections applied are indicated bellow:

  • Row 113 – ‘aria’ was corrected to ‘area’
  • Row 391 – the first phrase was changed from ‘more than meets the eye’ to ‘revealed complex details’
  • The conclusions section was rephrased from row 399 to 403: ‘Because the leaves of various medicinal plants are often used in extract preparation, a thorough and beforehand examination of these morphological and anatomical parameters could help determine if those plants are suited for pharmacological assessments.’

Reviewer 3 Report

After improvement the article is suitable for publication.

Author Response

Response to Reviewer 3 Comments

Thank you for your time and support!

This manuscript is a resubmission of an earlier submission. The following is a list of the peer review reports and author responses from that submission.

Round 1

Reviewer 1 Report

This manuscript is a study based on the morphological and anatomical analysis of the leaf of the Vinca species. This research is interesting in the morphology and anatomy of Vinca species. However, the title and conclusion are not appropriate. The chemical composition analysis did not work in this research. If the authors claim that the morphological and anatomical traits of the Vinca species are very different with regards to the chemical composition, the chemical composition of the Vinca species’ grown condition should be shown in this manuscript.

Figure 1 and 2

There are mistakes in the morphological terminology.

A guard cell is the cell surrounding the stomatal pore. The cells surrounding the stomatal pore are usually called “guard cells,” not “stomatal cells.”

The cells associating with the stoma are “subsidiary cells,” not “guard cell.”

Lines 101-102; If these facts in these lines are not the authors’ findings, some references should be added.

Figure 6

The computational estimation of the intercellular space of mesophyll was done using the pictures of Figure 5. The pictures of a and b in Figure 5 lack some part in the upper left. Therefore, the computational estimation data is incorrect.

Figure 7

Zoomed-out pictures are needed to confirm the pictures accurately.

Reviewer 2 Report

Dear authors,

I am providing several recommendations for this manuscript that you could accept, or refused, with the appropriate further argumentations.

Introduction: The goal is not clear enough, what is the aim of this research? Is it the study of morphology in relation of alkaloid content? Is it the assessment of its potential for use in the industry? Please, try to explain this in more specific way.
Also, please provide general principles how the morphology, in technical and physiological meaning, affects and relates to the alkaloid content. 

Results are presented well, the photography is very informative and clear. 

Materials and Methods: The statistical analyses are basic, but satisfactory for this study.

Discussion: The discussion section is mainly explanation of the results of the other authors, instead of putting into critical comparison of your results with the results of the others.
You should also explain how the morphology of the leaves affects, first, the contents of the alkaloids, and the second,  the extraction and the quantity of the alkaloids.

Conclusions: Try to specify, within the conclusions how the differences in the morphology affects the photosynthesis, and finally the content of alkaloids and final extracts. I mean, the conclusions in this way are also too general.

There are also several minor mistakes regarding the unusual, repetitious numbering of the literature sources within the manuscript, and missing the italics in the names of the plant species. 

For these comments, please see the document: there are several comments within the .pdf - they should be visible in the pdf readers like ''Pdf exchange viewer'', or ''Acrobat reader'', but probably in the other software, too.

Reviewer 3 Report

The study from Ciorȋƫǎ et al. is investigating of leaves anathomical and morphological changes  of four species of Vinca grown in  open space and in greenhouse. The leaves of V. minor and V. herbacea had a more organized anatomical aspect, compared to the leaves of V. major and V. major var. variegata. The article is well written and interesting. The epicuticular waxes and cuticles of Vinca leaf epidermises are described for the first time, along with a mesophyll airspace determination method and all these results provides a new perspective for other plant species  as well.

Reviewer's comments:

in the all text should be corrected botanical names of plants - it should be italicized 

please correct the first sentence of the introduction